# Neural Dynamics under Active Inference: Plausibility and Efficiency of Information Processing

**DOI:** 10.3390/e23040454

**Published:** 2021-04-12

**Authors:** Lancelot Da Costa, Thomas Parr, Biswa Sengupta, Karl Friston

**Affiliations:** 1Department of Mathematics, Imperial College London, London SW7 2AZ, UK; 2Wellcome Centre for Human Neuroimaging, Institute of Neurology, University College London, London WC1N 3BG, UK; thomas.parr.12@ucl.ac.uk (T.P.); b.sengupta@imperial.ac.uk (B.S.); k.friston@ucl.ac.uk (K.F.); 3Core Machine Learning Group, Zebra AI, London WC2H 8TJ, UK; 4Department of Bioengineering, Imperial College London, London SW7 2AZ, UK

**Keywords:** active inference, free energy principle, process theory, natural gradient descent, information geometry, variational Bayesian inference, Bayesian brain, self-organisation, metabolic efficiency, Fisher information length

## Abstract

Active inference is a normative framework for explaining behaviour under the free energy principle—a theory of self-organisation originating in neuroscience. It specifies neuronal dynamics for state-estimation in terms of a descent on (variational) free energy—a measure of the fit between an internal (generative) model and sensory observations. The free energy gradient is a prediction error—plausibly encoded in the average membrane potentials of neuronal populations. Conversely, the expected probability of a state can be expressed in terms of neuronal firing rates. We show that this is consistent with current models of neuronal dynamics and establish face validity by synthesising plausible electrophysiological responses. We then show that these neuronal dynamics approximate natural gradient descent, a well-known optimisation algorithm from information geometry that follows the steepest descent of the objective in information space. We compare the information length of belief updating in both schemes, a measure of the distance travelled in information space that has a direct interpretation in terms of metabolic cost. We show that neural dynamics under active inference are metabolically efficient and suggest that neural representations in biological agents may evolve by approximating steepest descent in information space towards the point of optimal inference.

## 1. Introduction

Active inference is a normative framework for explaining behaviour under the free energy principle, a theory of self-organisation originating in neuroscience [1,2,3,4] that characterises certain systems at steady-state as having the appearance of sentience [5,6]. Active inference describes agents’ behaviour as following the equations of motion of the free energy principle so as to remain at steady-state, interpreted as the agent’s goal [7].

Active inference describes organisms as inference engines. This assumes that organisms embody a generative model of their environment. The model encodes how the states external to the agent influence the agent’s sensations. Organisms infer their surrounding environment from sensory data by inverting the generative model through minimisation of variational free energy. This corresponds to performing approximate Bayesian inference (also known as variational Bayes) [2,3,8,9,10,11], a standard method in machine learning, or minimising the discrepancy between predictions and sensations [1,12]. Active inference unifies many existing theories of brain function [13,14], such as, for example, optimal control [15,16,17,18], the Bayesian brain hypothesis [19,20,21] and predictive coding [19,22,23,24]. It has been used to simulate a wide range of behaviours in neuropsychology, machine learning and robotics. These include planning and navigation [25,26,27,28,29,30,31], exploration [32,33,34,35,36], learning of self and others [37,38], concept learning [39,40,41], playing games [42,43,44], adapting to changing environments [45,46,47], active vision [48,49,50,51] and psychiatric illness [42,52,53,54,55]. Active inference agents show competitive or state-of-the-art performance in a wide variety of simulated environments [34,46,47,56].

The last decade has seen the development of a theory of how the brain might implement active inference consistently in neurobiology [1,7,49,57,58]. A number of predictions of this theory have been empirically validated, including the role of dopamine in decision-making [59,60] and free energy minimisation in exploration and choice behaviour [61,62]. This paper characterises the neural dynamics of this process theory from two complementary standpoints: (1) consistency with empirically driven models of neural population dynamics, and (2) the metabolic and computational efficiency of such dynamics.

Efficiency is an important aspect of neural processing in biological organisms [63,64,65,66,67,68] and an obvious desideratum for artificial agents. Efficiency of neural processing in biological agents is best seen in the efficient coding hypothesis by Horace Barlow [69,70,71], which has received much empirical support [65,66,72,73,74,75,76,77] (see, in particular, [64] for energetic efficiency) and has been a successful driver in computational neural modelling [75,78,79,80]. In brief, any process theory of brain function should exhibit reasonably efficient neural processing.

Active inference formalises perception as inferring the state of the world given sensory data through minimisation of variational free energy. This amounts to constantly optimising Bayesian beliefs about states of the outside world in relation to sensations. By beliefs, we mean probability distributions over states of the environment. These distributions score the extent to which an agent trusts that the environment is, or not, in this or another state. From an information theoretic viewpoint, a change of beliefs is a computation or a change in information encoded by the agent.

This belief updating has an associated energetic cost. Landauer famously observed that a change in information entails heat generation [81,82]. It follows that the energetics needed for a change in beliefs may be quantified by the change in information encoded by the agent over time (as the organism has to alter, e.g., synaptic weights, or restore transmembrane potentials) mathematically scored by the length of the path travelled by the agent’s beliefs in information space. There is a direct correspondence between the (Fisher) information length of a path and the energy consumed by travelling along that path [83,84]. To ensure metabolic and computational efficiency, an efficient belief updating algorithm should reach the free energy minimum (i.e., the point of optimal inference) via the shortest possible path on average. Furthermore, since an agent does not know the free energy minimum in advance, she must find it using only local information about the free energy landscape. This is a non-trivial problem. Understanding how biological agents solve it might not only improve our understanding of the brain, but also yield useful insights into mathematical optimisation and machine learning.

In the first part of this work, we show that the dynamics prescribed by active inference for state-estimation are consistent with current models of neural population dynamics. We then show that these dynamics approximate natural gradient descent on free energy, a well-known optimisation algorithm from information geometry that follows the steepest descent of the objective in information space [85]. This leads to short paths for belief updates as the free energy encountered in discrete state-estimation is convex (see Appendix A). These results show that active inference prescribes efficient and biologically plausible neural dynamics for state-estimation and suggest that neural representations may be collectively following the steepest descent in information space towards the point of optimal inference.

## 2. The Softmax Activation Function in Neural Population Dynamics

This section rehearses a basic yet fundamental feature of mean-field formulations of neural dynamics; namely, the average firing rate of a neural population follows a sigmoid function of the average membrane potential. It follows that, when considering multiple competing neural populations, relative firing rates can be expressed as a softmax function of average transmembrane potentials, as the softmax generalises the sigmoid in attributing probabilities to multivariate (resp. binary) inputs.

The sigmoid relationship between membrane potential and firing rate was originally derived by Wilson and Cowan [86], who showed that any unimodal distribution of thresholds within a neural population, whose individual neurons are modelled as a Heaviside response unit, results in a sigmoid activation function at the population level. This is because the population’s activation function equals a smoothing (i.e., a convolution) of the Heaviside function with the distribution of thresholds.

The assumption that the sigmoid arises from the distribution of thresholds in a neural population remained unchallenged for many years. However, the dispersion of neuronal thresholds is, quantitatively, much less important than the variance of neuronal membrane potential within populations [87]. Marreiros and colleagues showed that the sigmoid activation function can be more plausibly motivated by considering the variance of neuronal potentials within a population [88], which is generally modelled by a Gaussian distribution under the Laplace assumption in mean-field treatments of neural population dynamics [89]. Briefly, with a low variance on neuronal states, the sigmoid function that is obtained—as a convolution of the Heaviside function—has a steep slope, which means that the neural population, as a whole, fires selectively with respect to the mean membrane potential, and vice-versa. This important fact, which was verified experimentally using dynamic causal modelling [88,90], means that the variance of membrane potentials implicitly encodes the (inverse) precision of the information encoded within the population.

Currently, the sigmoid activation function is the most commonly used function to relate average transmembrane potential to average firing rate in mean-field formulations of neural population dynamics [91,92] and deep neural networks [93,94].

It follows that when considering multiple competing neural populations, the softmax of their average membrane potentials may be used to express their relative firing rates. This can be seen from the fact that the softmax function generalizes the sigmoid in attributing probabilities in multivariate (resp. bivariate) classification tasks (p. 198) [93]. We refer to Section 5.2, Chapters 3 and 6 in [95,96,97] for mathematical derivations of this generalisation.

## 3. Neural Dynamics of Perceptual Inference

Active inference formalises perception as inferring the state of the world given sensory data through the minimisation of variational free energy [7]. For state estimation on discrete state-space generative models (e.g., partially observable Markov decision processes [98]), the free energy gradient corresponds to a generalised a prediction error [7]. This means that to infer the states of their environment, biological agents reduce the discrepancy between their predictions of the environment and their observations, or maximise the mutual information between them [63].

Variational free energy is a function of approximate posterior beliefs *Q*,
$$\begin{array}{cc} F(Q) & ≜E_{Q(s)}\left[\ln Q(s)-\ln P(o,s)\right]\\ & =\underset{\geq 0}{\underset{︸}{D_{KL}\left[Q(s)\|P(s|o)\right]}}-\underset{\text{Log-evidence}}{\underset{︸}{\ln P(o)}}\\ & =\underset{\mathrm{Complexity}}{\underset{︸}{D_{KL}\left[Q(s)\|P(s)\right]}}-\underset{\mathrm{Accuracy}}{\underset{︸}{E_{Q(s)}\left[\ln P(o|s)\right]}} \end{array}$$
While *P* is the generative model: a probability distribution over hidden states (*s*) and observations (*o*) that encodes the causal relationships between them. Only past and present observations are directly accessible; hidden states can only be inferred. This means that the free energy depends on the sequence of available stimuli and is updated whenever a new observation is sampled. The symbol *E_Q_* means the expectation (i.e., the average) of its argument under the subscripted distribution. *D_KL_* is known as the Kullback–Leibler divergence or relative entropy [99,100,101] and is used as a non-negative measure of the discrepancy between two probability distributions. Note that this is not a measure of distance, as it is asymmetric. The second line here shows that minimising free energy amounts to approximating the posterior over hidden states, which are generally intractable to compute directly, with the approximate posterior. Exact Bayesian inference requires the approximate and true posterior to be exactly the same, at which point free energy becomes negative log model evidence (a.k.a., marginal likelihood). This explains why the (negative) free energy is sometimes referred to as an evidence lower bound (ELBO) in machine learning [93]. The final line shows a decomposition of the free energy into accuracy and complexity, underlying the need to find the most accurate and minimally complex explanation for sensory observations (c.f., Horace Barlow’s principle of minimum redundancy [69]).

When a biological organism represents some of its environment in terms of a finite number of possible states (e.g., the locations in space encoded by place cells), we can specify its belief by updating about the current state in peristimulus time. Discrete state-estimation is given as a (softmax) function of accumulated negative free energy gradients [57].
$$\begin{array}{l} \dot{v}\;=-\nabla_{s}F\\ s=\sigma (v) \end{array}$$

In this equation, *σ* is a softmax function and s represents the agent’s beliefs about states. (These are the parameters of a categorical distribution *Q* over states). Explicitly, **s** is a vector whose i-th component is the agent’s confidence (expressed as a probability) that is in the i-th state. The softmax function is the natural choice to map from free energy gradients to beliefs as the former turns out to be a logarithm [7] and the components of the latter must sum to one.

Just as neuronal dynamics involve translation from post-synaptic potentials to firing rates, these dynamics involve translating from a vector of real numbers (*v*), to a vector where components are bounded between zero and one (**s**). As such, we can interpret v as the voltage potential of neuronal populations, and **s** as representing their firing rates (since these are upper bounded thanks to neuronal refractory periods). Note the softmax function here plays the same role as in mean-field formulations; it translates average potentials to firing rates. On the one hand, this view is consistent with models of neuronal population dynamics. On the other hand, it confers post-hoc face validity, as it enables the synthesis of plausible local field potentials (see Figure 1) and a wide range of other electrophysiological responses, including repetition suppression, mismatch negativity, violation responses, place-cell activity, phase precession, theta-gamma coupling, and more [57].

The idea that state-estimation is expressed in terms of firing rates is well-established when the state-space constitutes an internal representation of space. This is the raison d’être of the study of place cells [102], grid cells [103] and head-direction cells [104,105], where the states inferred are physical locations in space. Primary afferent neurons in cats have also been shown to encode kinematic states of the hind limb [106,107,108]. Most notably, the seminal work of Hubel and Wiesel [109] showed the existence of neurons encoding orientation of visual stimuli. In short, the very existence of receptive fields in neuroscience speaks to a carving of the world into discrete states under an implicit discrete state generative model. While many of these studies focus on single neuron recordings, the arguments presented above are equally valid and generalise the case of “populations” comprising a single neuron.

In summary, the neuronal dynamics associated with state-estimation in active inference are consistent with mean-field models of neural population dynamics. This view is strengthened a posteriori, as this allows one to generate a wide range of plausible electrophysiological responses. Yet, further validation remains to be carried out by testing these electrophysiological responses empirically. We will return to this in the discussion.

## 4. A Primer on Information Geometry and Natural Gradient Descent

To assess the computational and metabolic efficiency of a belief trajectory, it becomes necessary to formalise the idea of “belief space”. These are well-studied structures in the field of information geometry [110,111,112], called statistical manifolds. In our case, these are (smooth) manifolds, where each point corresponds to a parameterisation of the probability distribution in consideration (see Figure 2). One is then licensed to talk about a change in beliefs as a trajectory on a statistical manifold.

Smooth statistical manifolds are naturally equipped with a different notion of distance, even though they may be subsets of Euclidean space. This is because the Euclidean distance measures the physical distance between points, while the information length measures distance in terms of the (accumulated) change in (Fisher) information (see Figure 2) along a path. The canonical choice of information length on a statistical manifold is associated with the Fisher information metric tensor **g** [113,114,115]. Technically, a metric tensor is a smoothly varying choice of symmetric, positive definite matrices at each point of the manifold. This enables computation of the length of paths as well as the distance between points, by measuring the length of the shortest path (see Appendix B). Mathematically, the Fisher information metric can be defined the Hessian of the KL divergence between two infinitesimally close distributions (see Appendix B). This means that the information length of a trajectory on a statistical manifold is given by accruing infinitesimally small changes in the KL divergence along it.

Amari’s natural gradient descent [85,116] is a well-known optimisation algorithm for finding the minimum of functions defined on statistical manifolds such as the variational free energy. It consists of preconditioning the standard gradient descent update rule with the inverse of the Fisher information metric tensor:$$\dot{s}=-\nabla_{s}F\;\to \;\dot{s}=-g^{-1}(s)\nabla_{s}F$$

Preconditioning by the inverse of **g** means that the natural gradient privileges directions of low information length. One can see this, since the directions of greatest (resp. smallest) information length are the eigenvectors of the highest (resp. lowest) eigenvalues of **g**.

In fact, Amari proved that the natural gradient follows the direction of steepest descent of the objective function in information space [85]. Technically, the natural gradient generalises gradient descent to functions defined on statistical manifolds. As the free energy for discrete state-estimation is convex (see Appendix A), this means that natural gradient descent will always converge to the free energy minimum via a short path.

To summarise, agents’ beliefs naturally evolve on a statistical manifold towards the point of optimal inference. These manifolds are equipped with a different notion of distance, namely, the information length. In the case of discrete state-estimation, beliefs evolve on the simplex towards the free energy minimum. Reaching the minimum with a short path translates into higher computational and metabolic efficiency. One scheme that achieves short paths for finding the minimum of the free energy is the natural gradient. In the next section, we will show that the neuronal dynamics entailed by active inference approximate natural gradient descent.

## 5. Active Inference Approximates Natural Gradient Descent

Discretising the (neuronal) dynamics prescribed by active inference and natural gradient descent give us the following state-estimation belief updates, respectively:$$s^{\left(t+1\right)}\leftarrow \sigma \left(\ln s^{\left(t\right)}-\epsilon \;\nabla_{s^{\left(t\right)}}F\right)\;\;\;s^{\left(t+1\right)}\leftarrow \frac{s^{\left(t\right)}-\epsilon \;g^{-1}(s^{\left(t\right)})\nabla_{s^{\left(t\right)}}F}{\sim }$$

In these equations, the logarithm is taken component-wise; ε is the step-size used in the discretisation and ~ denotes normalisation by the sum of the components to ensure that **s**^(*t*+1)^ lies on the simplex. (Natural gradient descent dynamics do not necessarily remain on the statistical manifold. This problem has been the object of numerous works that supplemented the natural gradient update with a projection step [117,118,119,120,121]. Here, we choose the simplest projection step to ensure that the result remains on the simplex: normalising with the sum of the components.)

These dynamics are approximately the same. We can equate them under a first order Taylor approximation of the exponential inside the softmax function:$$\begin{array}{ll} s^{\left(t+1\right)} & \leftarrow \sigma \left(\ln s^{\left(t\right)}-\epsilon \;\nabla_{s^{\left(t\right)}}F\right)\\ & =\frac{\exp\left[\ln s^{\left(t\right)}-\epsilon \;\nabla_{s^{\left(t\right)}}F\right]}{\sim }\\ & =\frac{s^{\left(t\right)}⊙\exp\left[-\epsilon \;\nabla_{s^{\left(t\right)}}F\right]}{\sim }\\ & ≃\frac{s^{\left(t\right)}⊙\left[\vec{1}-\epsilon \;\nabla_{s^{\left(t\right)}}F\right]}{\sim }\\ & =\frac{s^{\left(t\right)}-\epsilon \;\;s^{\left(t\right)}⊙\nabla_{s^{\left(t\right)}}F}{\sim }\\ & =\frac{s^{\left(t\right)}-\epsilon \;g^{-1}(s^{\left(t\right)})\nabla_{s^{\left(t\right)}}F}{\sim } \end{array}$$

The symbol ⊙ denotes the Hadamard product (elementwise multiplication). The last line follows since, on the simplex, the inverse of the Fisher information metric tensor is simply a diagonal matrix whose diagonal is **s** (see Appendix C).

Although these dynamics are approximately the same, this does not guarantee that the paths taken in the limit of infinitesimally small time steps (which correspond to continuous-time dynamics in physical and biological systems) will be the same. One can see this algebraically, since the number of time steps needed to reach the free energy minimum increases as the step-size decreases; thus, the difference between paths, which equals the sum of the differences at each timestep, is not guaranteed to converge to zero. Hence, it is necessary to verify that this approximation holds well in practice by analysing the discrepancy between paths using numerical simulations.

## 6. Numerical Simulations

In this section, we use numerical simulations of decision-making tasks under uncertainty to assess to what extent the inferential dynamics of active inference match the natural gradient descent on free energy. Our simulations compared the information length of the belief trajectories taken by both schemes, which reflects their computational and metabolic efficiency.

### 6.1. Methodology

We simulate two sequential decision-making tasks under uncertainty—a simple two-step maze task and a more complex abstract rule learning task. These tasks involve an agent equipped with a generative model (in our case, a partially observed Markov decision process [98]) and a repeated sequence of the following steps [7]:**Observation**: The agent receives an observation;**Perceptual inference:** The agent infers hidden states from observations by minimizing free energy;**Planning as inference:** The agent evaluates courses of action by evaluating their expected free energy, an objective which favours exploitative (goal-seeking) and explorative (ambiguity resolving) behaviour [7,33];**Decision-making:** The agent executes the action with the lowest expected free energy.**Learning:** The agent learns the generative model by accumulating data on hidden state transitions and on the likelihood mapping between states and outcomes.

#### 6.1.1. Overview of Tasks

The first paradigm simulates a rat in a T-Maze (see Figure 3). The T-Maze has a reward which is placed either in the right or left upper arm (in red). The bottom arm contains a cue that specifies the location of the reward. The rat’s initial location is the middle of the T-Maze. The initial conditions are specified such that the rat believes that it will claim the reward and avoid the unbaited upper arm. In the first three trials, the reward’s location is random. The optimal strategy then consists of collecting the cue (exploration) and using this information to collect the reward (exploitation). To achieve this, the rat must infer its location and the configuration of the maze, in addition to the route it will take. In addition, the agent has to learn that the cue provides reliable information that enables it to infer the reward’s location. After the first three trials, the reward’s location remains constant across trials. As there is no residual uncertainty to resolve by checking the cue, the optimal strategy then consists of directly collecting the reward. To do this, the agent has to learn the reward’s location.

The second paradigm is a simulation of abstract rule learning. In each trial, the agent is presented with a sequence of stimuli and has to guess the next in the sequence, upon which it receives positive or negative feedback. The stimuli are generated according to a rule unknown to the agent, so that there is always one unique correct answer. The goal of the agent is to provide correct answers while avoiding incorrect ones. To achieve its goal, the agent has to learn the rule generating the stimuli. In the first trials—and in the absence of knowledge—the agent guesses the missing stimulus. After each trial, the agent updates its prior beliefs based on past observations. After several trials, the agent has learned the contingencies of the task, whence it is able to predict the missing stimulus correctly and reliably. A complete specification of the generative model for this rule learning task may be found in [39]. The important consideration for our purposes is that this simulation is a more complex version of the T-Maze task, in the sense that it uses the same belief updating scheme and generative model. The key difference is that the generative model contains many more states: sixteen control states, 144 hidden states and 48 possible observations [39], Figure 3. This means that the simplex on which the free energy is defined has 143 dimensions.

#### 6.1.2. Two Schemes

We simulate each task with two different schemes: a standard active inference scheme, and a modified active inference scheme that differs only in that it performs perceptual inference using natural gradient descent. More explicitly, these agents, respectively, infer hidden states from observations in peristimulus time using the learning rule
$$s^{\left(t+1\right)}\leftarrow \sigma \left(\ln s^{\left(t\right)}-\epsilon \;\nabla_{s^{\left(t\right)}}F\right)\;\;\;s^{\left(t+1\right)}\leftarrow \frac{s^{\left(t\right)}-\epsilon \;g^{-1}(s^{\left(t\right)})\nabla_{s^{\left(t\right)}}F}{\sim },$$
with a common step size of ε = 0.25. Note that this optimization procedure is repeated at each step of the decision-making task, as a new observation updates the free energy landscape. Note also that the free energy changes in between trials due to the fact that the generative model is learned over time. The explicit form of the free energy for a partially observed Markov decision process is given in Appendix A.

All other aspects of belief updating (planning as inference, decision-making and learning) were conserved across schemes, following [7,39,57] and their publicly available software implementation (see Software note).

#### 6.1.3. Measuring Information Length

We simulate both tasks with 128 agents of each scheme and for 24 trials. For each agent, we measure the information length of perceptual inference by accumulating the information distance (Appendix E) between consecutive updates of the learning rule **s**^(*t*+1)^, **s**^(*t*)^.
$$d_{g}(s^{\left(t+1\right)},s^{\left(t\right)})=2\left\|\sqrt{s^{\left(t+1\right)}}-\sqrt{s^{\left(t\right)}}\right\|$$

### 6.2. Results

We plot the results of our numerical experiments in Figure 4.

We observe that, in the T-Maze task, standard active inference performs better than the active inference scheme modified with natural gradient descent. However, the opposite is true in the abstract rule learning task. Overall, the differences in information length between the two schemes are small compared to the overall information length of belief trajectories (less than 4% deviation on average in both tasks). This suggests that the difference between the two schemes is small.

The paradigm-specific difference in performance is due to differences in the generative model and the corresponding free energy landscapes. This means that it should be possible to find other paradigms where either active inference or the natural gradient performs better. Yet, any discrete state-estimation task will exhibit a quantitatively similar, convex-free energy landscape (Appendix A), which implies that the inference problem will be mathematically analogous. Therefore, we expect small performance differences between the two schemes for related sequential decision-making tasks (with an associated partially observed Markov decision process).

To summarise, our results suggest that state-estimation in active inference is a good approximation to natural gradient descent on free energy. Natural gradient descent follows the steepest descent (as defined in information space) on a convex free energy landscape (see Appendix A). This means that neural dynamics under active inference take short paths to the free energy minimum, which are computationally and metabolically efficient.

## 7. Discussion

In the first part of this paper, we showed that neural dynamics for discrete state-estimation under active inference are consistent with mean-field models’ neural population dynamics. This construct validity is further supported by the wide range of plausible electrophysiological responses that can be synthesised with active inference [57]. Yet, to fully endorse this view, the electrophysiological responses simulated during state-estimation need empirical validation. To do this, one would have to specify the generative model that a biological agent employs to represent a particular environment. This may be identified by comparing alternative hypothetical generative models with empirical choice behaviour and computing the relative evidence for each model (e.g., [123]). Once the appropriate generative model is found, one would need to compare the evidence for a few possible practical implementations of active inference, which come from various possible approximations to the free energy [27,124,125], each of which yields different belief updates and simulated electrophysiological responses. Note that, of possible approximations of the free energy, the marginal approximation which was used in our simulations currently stands as the most biologically plausible [124]. Finally, one would be able to assess the explanatory power of active inference in relation to empirical measurements and compare it with other process theories.

In the second part of this paper, we showed that the neuronal process theory associated with active inference approximates natural gradient descent for discrete state-estimation. (Note that this work does not treat continuous state-estimation, e.g., inferring the temperature in a room, which needs to be investigated separately). Given that the natural gradient follows the direction of steepest descent of the free energy in information space [85] and that the free energy landscape at hand is convex (see Appendix A), this ensures that agent’s beliefs reach the point of optimal inference via short trajectories in information space. This means that active inference entails neuronal dynamics that are both computationally and energetically efficient, an important feature of any reasonable process theory of the brain.

In the case of simulated (i.e., discretised) belief dynamics, active inference and natural gradient perform similarly on average. Performance is scored by the information length accrued during belief updating, which measures efficiency. In some cases, however, the belief trajectories taken by both schemes were significantly longer than the shortest path to the point of optimal inference. This is unsurprising since agents’ beliefs move myopically to the free energy minimum. In short, our analysis suggests that biological agents can perform natural gradient descent in a biologically plausible manner. From an engineering perspective, this means that we can relate variational message passing and belief propagation, two inferential algorithms based on free energy minimisation [124,126,127], with the natural gradient.

A general point is that the tools furnished by information geometry are ideally suited to characterise and visualise inference in biological organisms as well as scoring its efficiency. This paves the way for further applications of information geometry to analyse information processing in biological systems.

## 8. Conclusions

In the first part, we showed how a generic account of brain function provided by active inference is consistent with the more biophysically detailed and empirically driven mean-field models of neural population dynamics.

We then demonstrated that state estimation under active inference approximates natural gradient descent on free energy. This suggests that the beliefs about states of the world of biological agents evolve approximately according to a steepest descent on free energy. Since the free energy landscape for discrete state-estimation is convex, the trajectories taken to the point of optimal inference are short, which incurs minimal computational and metabolic cost. This demonstrates the efficiency of neural dynamics under active inference, an important feature of the brain as we know it. Further testing of active inference as a process theory of brain function should focus on extending the empirical evaluation of simulated neurophysiological responses.

## Figures and Tables

**Figure 1 entropy-23-00454-f001:**
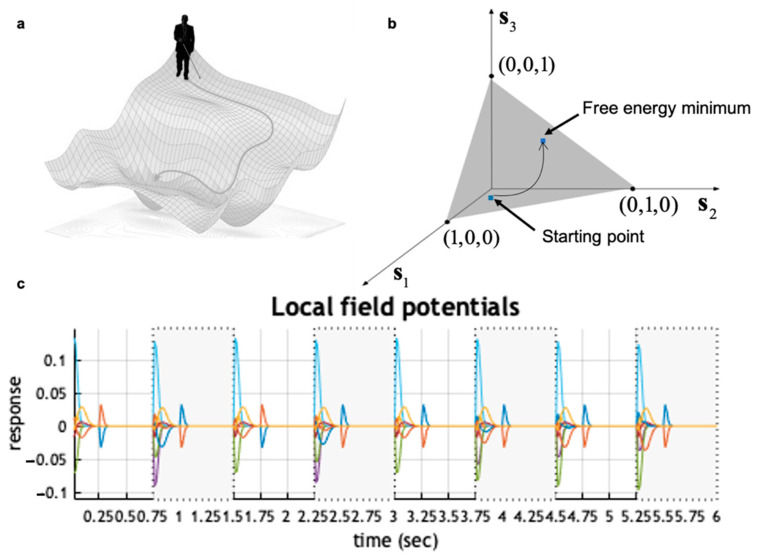
Active inference for state estimation (discrete state-space). Panel (**a**) summarises the problem of finding the minimum of a function (e.g., the free energy). One possibility would be taking the shortest path, which involves climbing up a hill; however, in the nescience of the minimum, a viable strategy consists of myopically taking the direction of steepest descent. In panel (**b**), we depict an example of a trajectory of an agent’s beliefs during the process of perception, which consists of updating beliefs about the states of the external world to reach the point of optimal inference (i.e., free energy minimum). In this example, the state-space comprises only three states (e.g., three different locations in a room). As they are probabilities over states, the components of s are non-negative and sum to one; hence, the agent’s beliefs naturally live on a triangle in three-dimensional space. Mathematically, this object is called a (two-dimensional) simplex. This constitutes the belief space, or information space, on which the free energy is defined. Technically, this object is a smooth statistical manifold, which corresponds to the set of parameters of a categorical distribution. To optimise metabolic and computational efficiency, agents must update their beliefs to reach the free energy minimum via the shortest possible path on this manifold. In panel (**c**), we exhibit simulated local field potentials that arise by interpreting the rate of change of *v* in terms of depolarisations, over a sequence of eight observations (e.g., saccadic eye-movements). As the rate of change is given by the free energy gradients, the decay of these local field potentials to zero coincides with the reaching of the free energy minimum (at which the gradient is zero by definition). Each colour expresses the mean voltage potential of a neural population, which encodes beliefs about one hidden state. These were obtained during the first eight trials of the T-Maze numerical simulation described in Section 6.1.1. Local field potentials accompanying the rule learning simulation—and for subsequent trials of the T-Maze task—can be found in [39,57], respectively. For more details on the generation of simulated electrophysiological responses, see [57].

**Figure 2 entropy-23-00454-f002:**
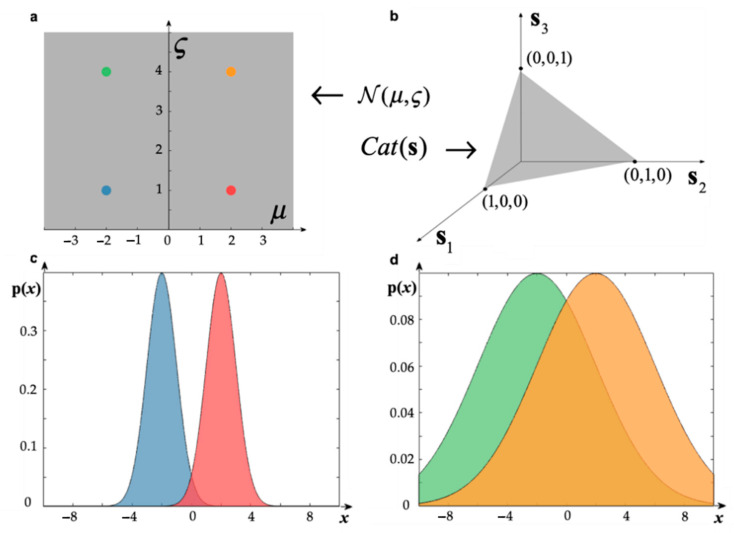
Statistical manifolds and information length. Panels (**a**,**b**) illustrate the statistical manifolds associated with two well-known probability distributions: the normal distribution and the categorical distribution, respectively. The statistical manifold associated with a probability distribution is the set of all possible parameters that it can take. For the univariate normal distribution, parameterised with mean *μ* and positive standard deviation ς, the associated statistical manifold is the upper half plane (panel **a**). For the categorical distribution, in the case of three possible states, the statistical manifold is the 2-dimensional simplex (panel **b**). More generally, in the case of n possible states, the statistical manifold of the categorical distribution is the set of all vectors with positive components that sum to one, i.e., the (n − 1)-dimensional simplex. This is a higher-dimensional version of the triangle or the tetrahedron. In panels (**c**,**d**) we illustrate why the Euclidean distance is ill-suited to measure the information distance between probability distributions. To show this, we selected four distributions that correspond to points on the statistical manifold of the normal distribution. One can see that the Euclidean distance between the modes of the red and the blue distributions is the same as that from the orange and the green; however, the difference in information of each respective pair is quite different. In panel (**c**), the two distributions correspond to two drastically different beliefs, since there is such little overlap; on the contrary, the beliefs in panel (**d**) are much more similar. This calls for a different notion of distance that measures the difference in (Fisher) information between distributions, namely, the information length.

**Figure 3 entropy-23-00454-f003:**
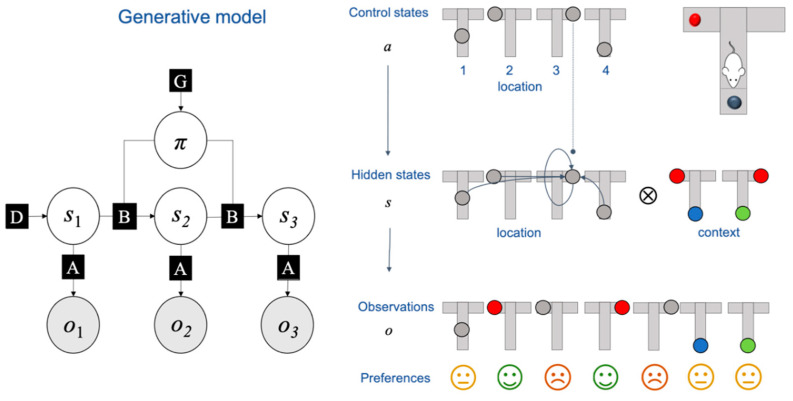
Generative model and T-Maze task. This figure describes the generative model used in the simulations and the specificities of the T-Maze task. The generative model is a partially observed Markov decision process (POMDP), a canonical model for sequential decision-making tasks under uncertainty. On the left-hand-side, the generative model is expressed as a factor graph [122]: the circles represent random variables while the squares represent factors. The shaded variables are observations, which the agent samples in discrete time. The unshaded variables are random variables that are inferred by the agent. These are either hidden states (*s*) or policies (π, i.e., action sequences), which need to be inferred from observations. The factors encode the relationships between the variables that they link. For example, D is a vector of non-negative components that sum to one that expresses the prior belief of being in this or another hidden state; B is a stochastic matrix that encodes the transition probabilities between hidden states given actions; similarly, A is a matrix that gives the probability of observations given states. See [7] for more details. Importantly, the factors can themselves be learned over time. On the right-hand side, we further unpack the generative model of the T-Maze task. The task is as pictured in the upper-right corner. The rat’s initial location is the centre of the T-Maze. At each time-step, the rat can choose between four control states (i.e., actions) that lead the rat to move to one of the four maze locations (middle, top-left, top-right or bottom). There are four hidden states that correspond to the location of the rat in the T-Maze and another two that correspond to the location of the reward (right arm or left arm), which leads to eight hidden states in total, meaning that the associated simplex on which the free energy is defined is seven-dimensional. There are seven possible observations that correspond to being in the middle of the T-Maze, in each one of the upper arms with (in red) or without the reward (in grey), and to collecting the cue, which reveals the location of the reward. The agent has a preference for each of these observations: the agent wants to claim the reward and avoid the unbaited arm. The agent is neutral w.r.t. all other observations. See [57] for more details of this paradigm.

**Figure 4 entropy-23-00454-f004:**
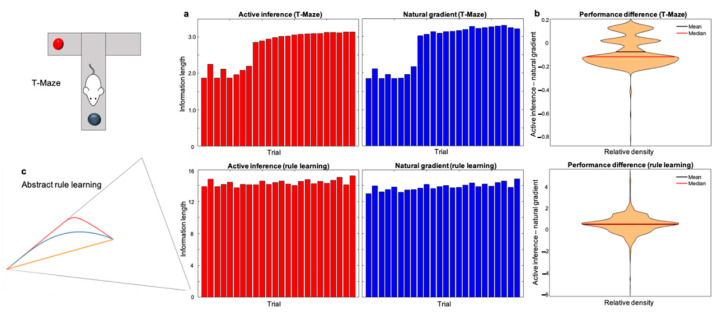
Information length of perceptual inference in active inference and natural gradient. The upper-half of the figure plots the results of the numerical experiments with T-Maze task, while the lower half the abstract rule learning task. In Panel (**a**), the histograms show the information length of perceptual inference in active inference (in red) and natural gradient (in blue). Specifically, this is (mean = 7075, median = 2.9776) for active inference, and (mean = 2.7793, median = 3.1163) for the natural gradient; in the rule learning task, the information lengths are (mean = 14.3573, median = 16.2099) for active inference and (mean = 13.8588, median = 16.1058) for the natural gradient. Information trajectories are longer in the abstract rule learning task, due to its increased complexity. In other words, the greater number of hidden states—in the generative model—mandate more belief updating. This means the model used for this high-dimensional task entails greater complexity in the technical sense: as belief updating moves posterior beliefs further from their priors. (Parenthetically, this means that the agents took much longer to update their beliefs). The reasons for systematic variation in information length across trials within tasks is that (1) the task configuration varied from trial to trial and (2) the generative models were themselves learned over trials. The sudden increase in information length in the T-Maze task at the ninth trial occurs when agents learn that the reward’s location is constant. As the reward’s location is unambiguous, belief trajectories are longer as beliefs about context move from the middle of the simplex to the vertex representing the reward’s location at *each* step of the paradigm. This means that we can improve the biological plausibility of the implementation by starting each belief update at the previous free energy minimum instead of an agnostic location in the middle of the simplex. This would avoid beliefs travelling to the same vertex multiple times in the same trial and hence would make belief trajectories *shorter* once the reward location is learned. This consideration does not affect our characterisation, which simply compares the information length given consistent starting and end points. As the context could be deduced only at the last step of the task, a similar sudden change in information length did not occur in the abstract rule learning task. However, the slight increase in information length across trials occurs for the same reason: learning the generative model made inferences increasingly precise. In Panel (**b**), the violin plots illustrate the difference in information length by subtracting the information length of the natural gradient from the information length of active inference. In the T-Maze paradigm, active inference mostly takes shorter paths, which is why the violin plot’s values are mostly negative (mean= −0.0718, median= −0.1146). However, we obtain the opposite pattern in the abstract rule learning task (mean = 0.4985, median= 0.5401). Panel (**c**) shows an example of the belief trajectory taken during state estimation in the abstract rule learning task. This was possible to include as the generative model includes a hidden state dimension with three possible alternatives [39], enabling a two-dimensional representation of the associated simplex. The red trajectory corresponds to active inference, the blue is natural gradient descent, and the orange is the shortest path to the free energy minimum (i.e., the geodesic, see Appendix D). This example is not representative of the average and was chosen for purely illustrative purposes as the trajectories are very distinct, lengthy and do not coincide with the geodesic. The fact that both schemes were significantly suboptimal in this example was unsurprising as beliefs evolve to the free energy minimum using only local information about the free energy landscape. Note that this suboptimality occurred only in a small minority of the trials considered here.

## Data Availability

This article has no additional data.

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
