# Peer review of "Neural Dynamics under Active Inference: Plausibility and Efficiency of Information Processing"

_entropy, 2021, doi:10.3390/e23040454_

Round 1
Reviewer 1 Report
Thank you for the clarifications. I have no further comments.
Reviewer 2 Report
I am satisfied with the revision.
Reviewer 3 Report
I find the manuscript to be significantly improved in this revision by the added sections on experiment design and the changes to the figures. The authors have addressed all of my points to my satisfaction and I recommend this manuscript for publication.This manuscript is a resubmission of an earlier submission. The following is a list of the peer review reports and author responses from that submission.
Round 1
Reviewer 1 Report
The manuscript is excellent from theoretical point of view.
The idea that the brain encodes free energy in terms of average membrane potential of a neural population is quite plausible and this might have important consequences in the field.
My only concern regards the limited simulation support of the the other theoretical aspect of the manuscript - that neural dynamics implements variational free energy minimization through natural gradient descent on the shortest information path. Simulation on a more complex free energy manifold would be more appealing.
As a minor comment, the authors might provide a more clear explanation how softmax generalizes the sigmoid function for (competing?) populations.
Reviewer 2 Report
I read this paper with interest as it brings together very interesting areas of optimization and brain science. My main issue was that I was not clear what the main contribution of the paper is - the algorithm appeared relatively standard and it was not quite clear to me how it serves to advance the application.
Reviewer 3 Report
The authors investigated the consistency between the neural dynamics under active inference and the mean-field models of neuronal population dynamics. The authors compared the information length between the active inference and the natural gradient descent in two different tasks during state-estimation. They suggested that the active inference approximates the natural gradient descent on free energy and concluded that the active inference is efficient and biologically plausible for state-estimation.
Major comments:
1) The authors tried to investigate the consistency between the mean-field model and the active inference. Although references for the numerical simulation details were provided, it is still necessary to include some essential information in this paper too. For example, information about the mean-field model simulation, the state sequence and the choices of the selective eight trials. Figure 1c did not indicate the different colour's meaning, are they representing different states or different mean-field models?
2) Figure 3b shows apparent longer information length after the first eight trials. Please explain why sucj differences in information length across task were only found in the T-maze task, but not in the abstract rule learning task? Do the eight selective trials in figure 1c come from the first eights trials in Figure 3b? Besides, the number of the y-axis in figure 3f is positive. However, the legend content indicates the natural gradient has a shorter information length than the active inference. Therefore, the number should be negative. Like the previous question, are the two schemes shown in figure 3 use the same simulation method in the reference [39,57]?
3) As in free moving behaviour, the rat may not choose at a fixed time point, would the neuronal population dynamics still be consistent with the active inference?
4) Would the information length still be similar for active inference and natural gradient descent in the incorrect state estimation or when the rule changes during learning?
5) The information length differences between natural gradient descent and active inference seems different when the scheme was changed. Would the similarity of information length depend on the paradigm? Can the active inference still approximate the natural gradient when some other learning scheme was applied (for example, hidden states are estimated based on the sequence of stimuli)?
Reviewer 4 Report
This is a very well-written manuscript. It provides important additions to the topic of free energy, and how it may be "implemented" in the brain. Further, it overall does a very good job at explaining the complicated underlying concepts in an understandable manner.
My main suggestions for improvement are regarding the figures: Figure 2 and 3 could use larger ticks and labels, in order to be readable. In 2c, 2d, and 3e left panel, axis labels should be added. Also, I recommend displaying the values for he performance difference in 3c and 3f with a violin plot instead of a box plot, this would convey more information and probably better support the conclusions. The figure legend for Figure 3 does not follow the order 3a to 3f, and perhaps could be revised to be more straightforward (or the arrangement/labeling of the panels could be altered). In particular, in lines 351-361 the text seems to jump from the results presented in 3c and 3f to the schematic of 3d, and back to discussing the results in a way I found confusing. Also, "atypical" is a bit vague as a term here.
Also, section 6 could perhaps also be a bit more detailed in presenting an overview what was done in the simulations and how the analyses were performed.
Line 262: The word "vanilla" is a bit vague, and could be replaced by something more clear.
